# Economic impact of a large-scale scabies upsurge on healthcare facilities in Rohingya refugee camps: A retrospective costing study

Charls Erik Halder[1]*, Md Abeed Hasan[1], James Charles Okello[1],
Sayed Sunny Uz Zaman[1], Julekha Tabassum Poly[1], Hamim Tassdik[1],
Dickson Wafula Barasa[1], Emmanuel Roba Soma[1], Md Farhad Hussain[3], U Maung Prue[1],
Sirajul Munir Khandokar[1], John Patrick Almedia[2], Jahangir Alam[1]

**1** Migration Health Division, International Organization for Migration (IOM), Cox's Bazar, Bangladesh,
**2** Regional Office for Asia and the Pacific, International Organization for Migration (IOM), Bangkok,
Thailand, **3** Cox's Bazar Medical College Hospital, Cox's Bazar, Bangladesh

* cehalder@iom.int

## Abstract

Scabies is one of the common infectious skin conditions globally, with a significantly high burden in hot and tropical countries, and resource-poor settings. Rohingya refugee camps in Cox's Bazar are one of the most protracted refugee crises in the world, sheltering approximately 1,143,096 refugees. While most existing literature focuses on mass drug administration (MDA) interventions or community-level estimates, the economic burden of scabies on health system is rarely studied. This is a retrospective costing study, where we used financial and epidemiological data from January 2021 to December 2024. Costing was done from the provider's perspective, focused on what International Organization for Migration (IOM) spent during the period as a health service provider. A combination of standard stepdown approach and micro-costing methods were used. Financial data were collected from the IOM health programme's annual budget and consumption reports. The study population included all individuals who were clinically diagnosed with scabies and received care at 35 IOM-supported health facilities in Cox's Bazar, Bangladesh. The overall estimated financial cost for IOM's scabies outbreak response was USD 2.12 million, with an annual average of USD 531,729. The average cost per scabies management ranged between USD 5.33 and USD 6.54. Drug costs accounted for 11.92% of the overall cost over 4 years. Of the total cost of USD 253,629.43 over 4 years, 79% was attributed to permethrin topical cream, which was used to manage an estimated 85% of the total managed cases. Scenario analyses demonstrates that the existing permethrin-based treatment preference is the most expensive treatment modality, compared with ivermectin-based treatment and mixed-treatment approaches. Although the average cost of treating scabies is relatively low, overall, the treatment cost for such a large population has a significant

**Data availability statement:** All data underlying the findings of this study and the analysis files are publicly available on Zenodo (DOI: https://doi.org/10.5281/zenodo.17697474). The repository contains fully anonymized, aggregated scabies case counts and the costing inputs used in the economic analysis for 2021–2024. No individual-level or potentially identifiable participant data are shared. Related datasets from the same scabies research series are available on Zenodo (DOIs: https://doi.org/10.5281/zenodo.17648621 and https://doi.org/10.5281/zenodo.17687717).

**Funding:** The authors received no specific funding for this work.

**Competing interests:** The authors have declared that no competing interests exist.

economic impact. This study found a substantial effect of MDA on reducing the financial burden on the healthcare system.

## Introduction

Scabies is one of the common skin conditions globally with significantly high burden in hot and tropical countries, resource poor settings and in areas with high population density. According to World Health Organization, the global incidence of scabies at any given time is more than 200 million, affecting 5–50 per cent of children in resource-poor areas [1]. Refugees and the displaced population possess a higher risk of scabies outbreak due to overcrowded living arrangements, insufficient access to water, sanitation, and laundering facility, limited access to healthcare services and shortage of essential medicines (e.g., scabicides) [2,3].

Scabies is a curable condition, effectively treatable with the topical application of permethrin or oral administration of ivermectin. Unlike non-communicable diseases or high burden communicable diseases, such as, Tuberculosis, HIV and Hepatitis, uncomplicated scabies does not require long-term treatment. The treatment cost of scabies, if considered as single unit, is relatively inexpensive. However, scabies has the high potentiality to cause massive outbreak, especially in overcrowded and refugee settings. Overlooking its collective economic burden in an outbreak setting may create misperception among the policy makers and public health professionals regarding its potential economic impact and the need for adequate planning of resources. For instance, a study in Fiji reported that although the average cost per presentation for scabies at primary healthcare was USD 17.7, the estimated annual healthcare costs of scabies and its complications was US$3.0 million equivalent to per capita cost of USD 3.3 [4].

The Rohingya refugee camps in Cox's Bazar is one of the largest protracted refugee crises in the world sheltering approximately, 1,143,096 refugees in 33 overcrowded refugee camps [5]. Overcrowded living arrangements, fragile shelters, inadequate water, sanitation, and hygiene (WASH) facilities together with heavy monsoons in both the refugee camps and neighboring host communities often result in outbreak of infectious diseases, diphtheria, measles, COVID-19, and acute watery diarrhea (AWD)/cholera. In the first study of our research series on scabies in this setting, we reported a massive outbreak that occurred between 2021 and 2024, with 384,852 cases and an overall attack rate of 5,562.59 cases per 10,000 population over the four-year period [6]. This high burden increased demand for clinical services and additional resources. Simultaneously, it contributed to disruptions in daily life, potentially hindering social interactions and economic activities within the refugee community. In response to the surge of scabies cases, a World Health Organization (WHO)-supported mass drug administration (MDA) campaign was implemented in the Rohingya refugee camps between November 2023 and January 2024. The campaign targeted approximately one million refugees and used oral ivermectin as the primary intervention. The MDA was coordinated with the Government of Bangladesh and implemented through health sector partners operating in the camps, including

humanitarian agencies. This large-scale intervention aimed to rapidly reduce transmission and prevent further escalation of the outbreak [7].

The high caseload of scabies in overcrowded humanitarian settings can place substantial strain on healthcare delivery. Outbreak response may increase demand for medicines, infection prevention and control supplies, staff time, consultation space, and other operational resources, especially in already overstrained health facilities [8]. Scabies also can lead to secondary bacterial infection, which could be complicated to life-threatening conditions, like acute glomer-ulonephritis and rheumatic fever, further efforts of healthcare workers and cost of the health system for managing the complications [9]. While most existing literature focuses on mass drug administration (MDA) interventions or community-level estimates, the economic burden of scabies on health facilities is rarely studied and documented [4,10]. However, understanding the direct healthcare costs of scabies is critical for planning interventions, allocating resources, and identifying cost-effective response strategies such as mass drug administration (MDA). To address this research gap, this study evaluated the financial impact of the massive upsurge of scabies on healthcare system. This research will assist health programme and policy makers to understand the financial implications of scabies upsurge and thus, facilitate proper planning for effective integration of scabies prevention and management within the essential health service model. This will also help examine potential impact and cost effectiveness of new prevention and management strategies or programme.

## Methodology

### Ethics statement

This study was conducted under a pre-approved protocol for secondary analysis of outbreak-related health and pro-gramme data. Ethical approval for the study protocol was obtained from the Ethical Review Board of Cox's Bazar Medical College Hospital (CoxMC/2023/017). Administrative permission to use outbreak-response data was obtained from the Office of the Civil Surgeon and the Refugee Relief and Repatriation Commissioner (RRRC), and internal approval was obtained from the International Organization for Migration (IOM) Migration Health Research Unit.

The data analyzed in this study were routinely collected as part of public health outbreak response and health ser-vice delivery activities between 2021 and 2024, and were subsequently used for research purposes under the approved secondary-use protocol. Epidemiological case data were extracted from routine outbreak surveillance databases, while financial data were obtained from programme budget and expenditure records. The research team accessed only ano-nymized, aggregated numerical data and did not have access to identifiable individual-level information at any stage of the study. Given the retrospective nature of the study and the exclusive use of anonymized, aggregated data, individual informed consent was not required.

### Study setting and population

The study was implemented in Rohingya refugee camps in Ukhiya and Teknaf in Cox's Bazar. The study population included all individuals who were clinically diagnosed with scabies and received care at 35 IOM-supported health facili-ties in Ukhiya and Teknaf, Bangladesh, between 1st January 2021 and 31st December 2024. Scabies caused a massive upsurge during this period in the refugee camps that prompted a substantial response from IOM. Therefore, these camps were selected for this study to get an in-depth information of the outbreak.

All scabies cases were diagnosed clinically at 35 IOM supported facilities in Ukhiya and Teknaf. MSF clinical guidelines – diagnosis and treatment manual was used for diagnosis and clinical management of the cases. Topical 5% permethrin cream was mainly used for treatment for uncomplicated scabies, while oral ivermectin and benzyl benzoate were used in selected scenarios according to clinical indication and availability. Cases with secondary bacterial infection received addi-tional treatment such as flucloxacillin where indicated, and cetirizine was used for symptomatic relief of itching.

## Study design

This is a retrospective costing study, where we have used financial and epidemiological data from January 2021 to December 2024. For this retrospective analysis, financial data and de-identified, aggregated epidemiological data were accessed for research purposes between 05/01/2025 and 20/02/2025. Costing was done from provider perspective focused on what IOM spent during the period as health service provider. A combination of standard stepdown accounting and micro-costing methods were used [11,12].

Micro-costing (or ingredient-based costing) was calculated direct cost per-case by itemizing resources used per patient [13]. In this study, we estimated the cost per patient for medicines, laboratory testing, and infection prevention and control supplies used in the management of both uncomplicated scabies and infected scabies, including treatment related to secondary bacterial infections where applicable.

Step-down costing method was used to estimate costing related to capitals, coordination, supervision, operation and maintenance, capacity building, risk communication and community engagement, volunteer payment and salaries and benefits of healthcare workers, meaning that total costs and resources utilized by the whole health programme of IOM during the period was systematically and proportionately distributed to the "specific service unit". The specific service unit refers, hereby, the scabies consultation and treatment service [14].

The study did not take account of the economic costs (value of resources that could have been productively used elsewhere) such as the value of volunteer time, donated goods, shared infrastructure in the scabies management. All staff and volunteers involved were salaried under the program, and all supplies, equipment, and logistics were directly financed by IOM.

## Data collection and management

Financial data for the study period 2021–2024 were collected from the IOM health programme's annual budget and consumption reports. A cost recording and data analysis tool was developed to record and analyze all relevant costs associated with scabies response (S1 File). Field visits to health facilities and meetings with healthcare workers and medical logisticians were made to develop the cost recording and data analysis tool. All costs were categorized by a) capital investment and b) recurrent costs. Financial and epidemiological data were accessed for research purposes between 05/01/2025 and 20/02/2025 as part of this retrospective analysis. The research team accessed only de-identified, aggregated data and did not have access to any personally identifiable information during or after data extraction and analysis.

For shared resources, for instance, human resources (healthcare workers and volunteers), coordination, operations and maintenance, and laboratory costs, annual costs were estimated based on the total expenditure reported in the financial documents and relevant procurement orders of the programme. The total costs were then proportionally allocated for scabies using the morbidity burden of scabies in comparison to the total outpatient consultations (Table 1). Scabies response specific investment includes capacity building of healthcare workers and community health workers and development and printing of information, education and communication materials.

**Table 1. Year-wise case load and proportional morbidity of scabies.**

|  | 2021 | 2022 | 2023 | 2024 |
|---|---|---|---|---|
| Scabies | 32,046 | 140,590 | 150,470 | 61,746 |
| Infected scabies | 2,083 | 7,348 | 10,577 | 2,312 |
| Proportional morbidity (over total consultation) | 0.04 | 0.11 | 0.12 | 0.05 |
| **Total outpatient consultations (all causes)** | 893,925 | 1,302,029 | 1,275,773 | 1,126,522 |

Note: Proportional morbidity was calculated as scabies consultations divided by total outpatient consultations (all causes) for each year.

Capital costs included biomedical and clinical equipment used for scabies patient management, and costs related to health facility construction/renovation. The costs were annualized using the formula below, assuming the lifespan of construction for 20 years and clinical equipment for 3 years (Creese and Parker, 1994; MSF).

$$\text{Annual Depreciation Cost} \ = \ \text{Useful Life (in years)/Purchase Cost of Asset}$$

Coordination costs include salaries of programme staff and office costs at Cox's Bazar and IOM standard 7% overhead cost over the total operation and staff cost. Supervision cost includes salaries of supervisory staff, e.g., clinical supervisors, health facility in-charges and national officers. The cost of drugs was calculated on per case basis by considering the number of drugs given per patient according to standard clinical protocol and procurement prices as specified in the procurement order.

Data management and cost calculations were performed using Microsoft Excel (Microsoft Corporation, USA). The following assumptions were used to estimate drug and laboratory costs (Table 2). The percentage of case load utilizing a specific medicine was determined by reviewing the consumption report. A 3% discounting rate was used to factor for inflation across year, aligning with WHO guidance [15].

Operation and maintenance costs for health facilities included repair and maintenance, rental of land and spaces, generator and utilities, fuel, vehicle maintenance and security cost. risk communication and community engagement (RCCE) include costing for organizing awareness raising events, focus group discussions, courtyard sessions, and information, education and communication materials dedicated for scabies prevention and management. Salaries and benefits are counted for healthcare workers engaged in providing outpatient services, including triage, consultation, nursing and medication/pharmacy. Remuneration was costed for community health workers (CHW) engaged in RCCE and facility-based volunteers engaged in triage, counselling and crowd control.

**Table 2. Basis and assumptions of calculating costing of drugs and laboratory tests.**

| Category | Type of medicine | % of case load (based on consumption report) | # of items per case | Unit price (BDT) |
|---|---|---|---|---|
| **Drugs** | Permethrin 5% Ointment | 85.00% | 2 | 35 |
| | Ivermectin tab | 10.00% | 2 | 10 |
| | Flucloxacillin 500mg cap | 5.00% | 20 | 4.8 |
| | Flucloxacillin 250mg cap | 2.00% | 20 | 3.1 |
| | Flucloxacillin syp 125mg/5ml | 3.00% | 1 | 50 |
| | Cetrizine tab | 64.00% | 5 | 0.82 |
| | Cetrizine syp | 36.00% | 1 | 9.2 |
| | Benzyl Benzoate 25% ointment | 5.00% | 0.5 | 36 |
| **Laboratory tests** | Renal function test, 2% of Infected case, 150 test per unit | 0.12% | 0.06 | 2,960 |
| **IPC supplies** | Gloves | 100% | 2 | |
| | Mask | 5% | 1 | |
| | Hand Sanitizer (1 bottle per 1000 case) | 100% | 0.001 | |

Note: Unit prices are in BDT (Bangladeshi Taka). Permethrin 5% cream is commonly supplied as a 30 g tube; standard instruction was two applications, one week apart, and two tubes were issued per patient visit to support household contact treatment. Flucloxacillin syrup (125 mg/5 mL) is commonly supplied as a 100 mL bottle. Cetirizine tablets are 10 mg per tablet; cetirizine syrup (5 mg/5 mL) is commonly supplied as a 60 mL bottle. IPC unit prices were calculated per item based on procurement unit costs; for reference, common retail unit prices in Bangladesh include gloves (BDT 25 per pair), surgical mask (BDT 5 per piece), and hand rub/sanitizer 500 mL (about BDT 205 per bottle)

## Scenario analyses

A scenario-based costing analysis was conducted to measure the impact of different choice of treatment on the costing of scabies management at health facilities to find out the best approach for cost-efficient treatment of scabies. Scenarios were developed by the public health experts of IOM and based on different proportions of supply availability of permethrin, ivermectin and benzyl benzoate (Table 3). All other capital and recurrent costs were held constant to isolate the effect of different level of supply availability on per-case patient costing.

In current practice (scenario 1), each patient receives two tubes of permethrin at their first visit in the clinic and based on discussion with clinicians, roughly two family members per household visit the clinics. Thus, a family of four members of different age groups roughly receives four permethrin tubes, which are shared across the members for two applications in one week apart. Ivermectin tablets are reserved only for patients with secondary bacterial infections. In this study, complicated cases referred to scabies cases with secondary bacterial infection or other clinical presentations requiring treatment beyond standard topical scabicidal therapy.

In scenarios 2 and 3, ivermectin was modelled as a treatment option for uncomplicated scabies cases as well as infected cases, rather than being reserved only for infected cases as in current practice. To account for household contacts treatment, one patient was allocated on average 5 ivermectin tablets of 6 mg with the same assumption that approximately two members per family accessed care at the health facility.

## Result

Table 4 demonstrates the annual cost breakdown of scabies outbreak response from the years 2021–2024 at IOM-supported health facilities in the Rohingya refugee camps and host communities in Cox's Bazar. The overall estimated financial cost for IOM's scabies outbreak response through its health facilities was USD 2.12 million with an annual average of USD 531,729.

Aligning with the rise of scabies cases from 2021 to 2023, annual cost of scabies-related health service delivery increased four-fold from USD 209,594 in 2021 to USD 812,854 in 2023. Following the MDA campaign, the cost sharply decreased to USD 354,858.76 in 2024. The outbreak peak period (2022–2023) had an average annual cost of USD 781,232.35, compared with USD 209,594.27 in the pre-surge year (2021) and USD 354,858.76 in the post-MDA year (2024). The average cost per scabies management ranged between USD 5.33 to USD 6.54, with the highest in 2021 and lowest in 2022.

Healthcare worker remuneration made up 19.89% of the overall cost, consistently sharing the largest portion of the annual cost across the years, ranging from USD 35,226 (in 2021) to USD 165,404.15 (in 2023). In addition to this cost, $186,150 was required for the remuneration of community health workers and health-facility-based volunteers, which is another 8 – 10% of the annual cost.

**Table 3. Different planning scenario of cost analyses based on different proportion of supply availability of scabicidal agents.**

|  | Scenario I (Current) | Scenario II | Scenario III |
|---|---|---|---|
|  | Permethrin-based | Ivermectin-based | Equal permethrin & Ivermectin |
| **Permethrin** | 85% | 15%* | 47.5% |
| **Ivermectin** | 10%** | 80% | 47.5% |
| **Benzyl Benzoate*** | 5% | 5% | 5% |

* Required to manage children <15 kg and pregnant women

** Required to manage complicated cases

*** Kept constant throughout as back up to support in case of stock-out of both permethrin and ivermectin

**Table 4. Estimated retrospective costing analysis for scabies outbreak response (2021 – 2024).**

| COST CATEGORIES | Type of cost | Costing approach | 2021 USD | 2021 % of total | 2022 USD | 2022 % of total | 2023 USD | 2023 % of total | 2024 USD | 2024 % of total | 4 YEARS TOTAL USD | 4 YEARS TOTAL % of total |
|---|---|---|---|---|---|---|---|---|---|---|---|---|
| A. CAPITAL COSTS | Biomedical & Clinical equipment | Step down | 4,587.71 | 2.19% | 20,126.88 | 2.68% | 21,541.30 | 2.65% | 8,839.56 | 2.49% | 55,095.46 | 2.59% |
| | Health facility construction/renovation | Step down | 16,687.54 | 7.96% | 50,263.58 | 6.71% | 54,903.02 | 6.75% | 25,514.60 | 7.19% | 147,368.75 | 6.93% |
| SUB TOTAL | | | 21,275.25 | 10.15% | 70,390.47 | 9.39% | 76,444.32 | 9.40% | 34,354.17 | 9.68% | 202,464.21 | 9.52% |
| B. RECURRENT COSTS | Supervision | Step down | 15,755.52 | 7.52% | 47,456.30 | 6.33% | 51,836.61 | 6.38% | 24,089.58 | 6.79% | 139,138.01 | 6.54% |
| | Healthcare workers remuneration | Micro-costing | 35,226.57 | 16.81% | 154,543.56 | 20.62% | 165,404.15 | 20.35% | 67,874.29 | 19.13% | 423,048.56 | 19.89% |
| | Volunteer remuneration (CHWs and facility-based volunteers) | Step down | 21,079.00 | 10.06% | 63,490.84 | 8.47% | 69,351.18 | 8.53% | 32,228.97 | 9.08% | 186,150.00 | 8.75% |
| | Drugs | Micro-costing | 21,119.31 | 10.08% | 92,653.18 | 12.36% | 99,164.41 | 12.20% | 40,692.53 | 11.47% | 253,629.43 | 11.92% |
| | IPC supplies | Micro-costing | 8,958.46 | 4.27% | 39,301.93 | 5.24% | 42,063.89 | 5.17% | 17,261.09 | 4.86% | 107,585.38 | 5.06% |
| | Laboratory consumables | Micro-costing | 1,922.76 | 0.92% | 8,435.40 | 1.13% | 9,028.20 | 1.11% | 3,704.76 | 1.04% | 23,091.12 | 1.09% |
| | Health facility – operation and maintenance (including repair, rental, utilities, vehicle, fuel) | Step down | 30,973.23 | 14.78% | 93,292.67 | 12.45% | 101,903.77 | 12.54% | 47,356.86 | 13.35% | 273,526.53 | 12.86% |
| | Capacity building | Step down | 2,000.00 | 0.95% | 4,120.00 | 0.55% | 4,243.60 | 0.52% | 4,370.91 | 1.23% | 14,734.51 | 0.69% |
| | Risk Communication and Community Engagement (events, sessions, IEC materials) | Step down | 2,916.26 | 1.39% | 2,938.94 | 0.39% | 5,832.52 | 0.72% | 1,035.11 | 0.29% | 12,722.83 | 0.60% |
| SUB TOTAL | | | 139,951.11 | 66.77% | 506,232.82 | 67.53% | 548,828.32 | 67.52% | 238,614.11 | 67.24% | 1,433,626.36 | 67.40% |
| SERVICE DELIVERY AND OPERATIONAL EXPENSES (Capital Costs + Recurrent Costs) | | | 161,226.36 | 76.92% | 576,623.29 | 76.92% | 625,272.64 | 76.92% | 272,968.28 | 76.92% | 1,636,090.56 | 76.92% |
| C. COORDINATION COST | Programme staff, office costs and overhead costs (23.08% of service delivery and operational expenses) | Step down | 48,367.91 | 23.08% | 172,986.99 | 23.08% | 187,581.79 | 23.08% | 81,890.48 | 23.08% | 490,827.17 | 23.08% |
| TOTAL ANNUAL COST | | | 209,594.27 | 100.00% | 749,610.27 | 100.00% | 812,854.43 | 100.00% | 354,858.76 | 100.00% | 2,126,917.73 | 100.00% |
| COST PER CASE | | | 6.54 | | 5.33 | | $5.40 | | 5.75 | | $5.53 | |

The drug costs consisted 11.92% of the overall cost in 4 years. It increased from USD 19,906.98 (11.37%) in 2021 to a peak of USD 99,164.41 (14.08%) in 2023, before dropping to USD 41,913.31 (14.12%) in 2024. The cost recording and data analysis tool used for the retrospective costing calculations is provided in the Supporting Information (see S1 File). Of the total cost of USD 253,629.43 in 4 years, 79% of cost was for permethrin topical cream being used for managing an estimated 85% of the total managed cases. On the other hand, tablet ivermectin only accounted for 2.58% of the drug cost, used for management of an estimated 10% of the caseload.

Additionally, an estimated USD 107,585 was utilized for infection prevention and control and USD 23,091 was utilized for laboratory consumables, making up 5% and 1% of the overall cost in 4 years, respectively.

Operational and maintenance expenses for health facilities accounted for an estimated 12–15% of the annual cost. This proportion was calculated based on the contribution of scabies to the total number of consultations managed by these facilities.

Investment in risk communication and community engagement and capacity building were relatively low throughout the period, each category sharing less than 1% of the overall annual budget.

A comparative analysis of different scenarios of proportions of patients treated with topical permethrin and oral ivermectin.

It demonstrates that existing permethrin-based treatment preference is most expensive treatment modality (Table 5 and Fig 1). Using ivermectin as the preferred treatment (scenario 2) drug costs can be reduced to nearly 83.5% of a permethrin-based approach, saving 2.55% of the overall budget. In case both drugs are utilized in equal proportion (scenario 3), drug costs drop to 94.12% of the current practice with an overall 1% total cost reduction.

**Table 5. Scenario analysis.**

|  | SCENARIO-1 | Ratio | SCENARIO-2 | Ratio | SCENARIO-3 | Ratio |
|---|---|---|---|---|---|---|
| **DRUG COST** | USD 253,629.43 | 1 | USD 211,983.90 | 83.58% | USD 238,718.36 | 94.12% |
| **NON-DRUG COST** | USD 1,873,288.30 | 1 | USD 1,860,794.64 | 99.33% | USD 1,868,814.98 | 99.76% |
| **TOTAL COST** | USD 2,126,917.73 | 1 | USD 2,072,778.54 | 97.45% | USD 2,107,533.34 | 99.09% |

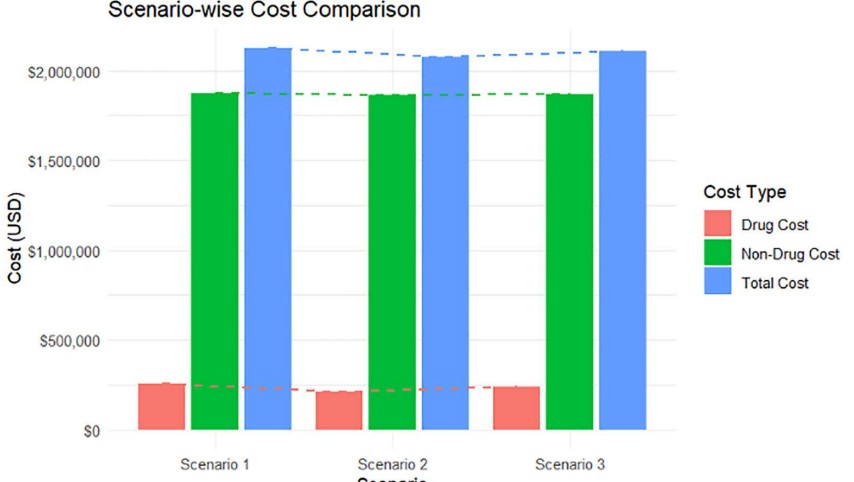

**Fig 1. Scenario-wise cost comparison of scabies response – Comparison of total, drug, and non-drug costs across three costing scenarios.** Drug costs represent expenditures on scabicidal medicines, while non-drug costs include human resources, operations, logistics, coordination, and other programme-related expenditures. All costs are presented in United States dollars (USD).

## Discussion

To our knowledge, this is the first study in a humanitarian and low-resource setting which explored the economic impact of scabies in health system. The study revealed a high economic impact of scabies on the health system valuing USD 2.12 million over four years with an average treatment cost of USD 5.53 per case. The study also found significant impact of mass drug administration on reducing economic burden on health system.

The average cost per scabies management ranged between USD 5.33 to USD 6.54. Since there was no other cost-analysis study available for scabies at humanitarian crises and massive outbreak settings, we could not compare our results with similar settings. However, the cost in our refugee setting was lower than the hospital setting in India and the primary care setting in Fiji, where the estimated average cost per patient were USD 12.20 and USD 17.7, respectively. The cost per patient also seems lower than other health burdens, such as tuberculosis, HIV, hepatitis, and non-communicable diseases [4,16]. However, if we consider the massive scale of the outbreak, the overall financial cost of the scabies upsurge had significant impact on the health system in the low resource setting.

We have found that the estimated financial cost for the IOM's scabies outbreak response was USD 2.12 million from 2021 to 2024 with an annual average of USD 531,729. This annual average is 3.33% of the overall funding appeal made by IOM for in health programme in 2024, indicating a significant economic impact of scabies in health response [17]. While IOM covers only one-fourth of the health facilities in the refugee camps, extrapolating it to sector wide response may cost around USD 2.1 million for full coverage of the catchment population in a year, which is around 2.5% of the appeal made by the health sector under the Joint Response Plan in 2025 [18]. Our finding coincides with the findings of the country-level recent study of Fiji, where the researchers found although a cost measure of single case of scabies is low, extrapolating the finding across the country, the estimated cost of scabies on the Fijian health system was ~ USD 3.0 million per year, equivalent to USD 3.3 per capita [4].

Our study found that annual cost for scabies outbreak response was drastically reduced in 2024 followed by the mass drug administration (MDA) campaign (November 2023 – January 2024), which was 43% of the previous year. Our previous study found that MDA contributed to a significant reduction of scabies cases, which sustained for six months followed by an upward shifting [6]. Aligning with this, our cost-analysis also suggests MDA contributes to significant reduction of economic burden in health systems in high prevalence context. While the estimated cost saving following MDA in IOM facilities was USD 457,996, extrapolating it to sector-wide response may lead to a saving of approximately USD 2.0 million with an investment of USD 1M required for the campaign [7]. The effectiveness and cost-efficiency of Mass Drug Administration have also been noted in other settings, like Fiji and Ethiopia [10,19]. In overcrowded and high burden settings, where the risk factors cannot be addressed properly, by treating the entire community at a time, MDA allows to substantially reduce the scabies prevalence and associated costs. The effectiveness of MDA will only be applicable until the scabies prevalence remains >10% of the community members [19].

We have studied three scenarios to explore the most cost-effective treatment regimen for scabies with first scenario demonstrating the existing practice of permethrin-based treatment, second scenario with ivermectin-based treatment and a third scenario where equal proportion of cases are managed by ivermectin and permethrin. Our analysis found that having ivermectin as preferred treatment (scenario 2) can reduce 16.5% of the drug cost and 2.55% over the overall scabies-related direct and indirect cost. a permethrin-based approach. When both drugs are utilized in equal proportion, 5.88% of drug costs can be reduced, while there will only be a reduction of 1% of the total cost. Topical permethrin and oral ivermectin are both effective and recommended treatment regimen for scabies, although ivermectin has contraindications in case of pregnancy and infants [1,14]. A few studies suggest that topical permethrin is clinically more effective than oral ivermectin [8]. In India, it was found that although the total cost of treatment with oral ivermectin was lower than topical permethrin, cost for relieving itching and transportation was higher than permethrin [20]. In endemic and high-burden setting, ivermectin can be more effective if there remains significant concern on the compliance [19]. Therefore, the decision of preferred treatment should not only be

decided based on this economic analysis presented here, but also consideration of endemicity, compliance, clinical outcome and response time.

This was a retrospective costing analysis, therefore, in case unavailability of exact data, the researchers had to estimate the nearest possible cost based on the discussion on the relevant stakeholders. Another limitation was our study did not measure the economic impact of scabies from the patient and social perspective. For instance, this massive scabies outbreak may have impact on the quality of life and result in social exclusion and stigma and absenteeism at school and workplace [1]. There could also be costs related to scabies related to productivity loss, time of caregiving and travel. Therefore, while further research is warranted to understand the overall impact of scabies on health system and community, prevention and control strategies, including mass drug administration, should be imposed to reduce the burden of scabies on health system and people's lives [1].

## Conclusion

Scabies has a high impact on health-related quality of life. Our results indicated that, although the average cost of treating scabies is relatively low, overall, the treatment cost for such a large population has a significant economic impact. This study found a substantial effect of mass drug administration on reducing the financial burden on the healthcare system. Strengthening mass drug administration and reinforcing preventive measures is crucial to prevent such an upsurge, especially in low-resourced settings like that of the Rohingya refugee camp.

## Supporting information

**S1 File. Detailed cost recording and data analysis tool used for retrospective costing calculations.**
(XLSX)

## Acknowledgments

CEH conceptualized and designed the study, provided overall methodological guidance, and led the drafting and critical revision of the manuscript. MAH assisted in refining the study design and drafting the manuscript. SSUZ, JTP, HT supported field coordination, MAH, JA assisted supporting the data collection processes. MAH supported with climatic data triangulation. DWB, ERS and JCO contributed to the critical review and refinement of the manuscript. MFH, ERS, UMP and JCO provided strategic oversight and ensured alignment of the study within broader programmatic and policy frameworks. All authors reviewed and approved the final version of the manuscript. CEH is the corresponding author and JCO is the guarantor of the study.

## Author contributions

**Conceptualization:** Charls Erik Halder.

**Data curation:** Charls Erik Halder, Md Abeed Hasan, Sayed Sunny Uz Zaman, Hamim Tassdik, Sirajul Munir Khandokar, Jahangir Alam.

**Formal analysis:** Charls Erik Halder, Md Abeed Hasan, Julekha Tabassum Poly, Sirajul Munir Khandokar.

**Investigation:** Charls Erik Halder, Sayed Sunny Uz Zaman, Julekha Tabassum Poly.

**Methodology:** Charls Erik Halder, Md Farhad Hussain.

**Project administration:** Charls Erik Halder, James Charles Okello, Dickson Wafula Barasa, U Maung Prue.

**Resources:** Charls Erik Halder, Md Abeed Hasan, James Charles Okello, Sayed Sunny Uz Zaman, Emmanuel Roba Soma, Jahangir Alam.

**Software:** Charls Erik Halder.

**Supervision:** Charls Erik Halder, James Charles Okello, Dickson Wafula Barasa, Emmanuel Roba Soma, U Maung Prue.

**Validation:** Charls Erik Halder, James Charles Okello, Emmanuel Roba Soma.

**Visualization:** Charls Erik Halder.

**Writing – original draft:** Charls Erik Halder.

**Writing – review & editing:** Charls Erik Halder, Md Abeed Hasan, James Charles Okello, Dickson Wafula Barasa, Emmanuel Roba Soma, Md Farhad Hussain, U Maung Prue, Sirajul Munir Khandokar, John Patrick Almedia.

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
