## [Decision Letter · Decision Letter 0]

25 Feb 2026

PGPH-D-25-03807

Economic Impact of a Large-Scale Scabies Upsurge on Healthcare Facilities in Rohingya Refugee Camps: A Retrospective Costing Study

Dear Dr. Halder,

Thank you for submitting your manuscript to PLOS Global Public Health. After careful consideration, we feel that it has merit but does not fully meet PLOS Global Public Health’s publication criteria as it currently stands. Therefore, we invite you to submit a revised version of the manuscript that addresses the points raised during the review process.

Please note that we have only been able to secure a single reviewer to assess your manuscript. We are issuing a decision on your manuscript at this point to prevent further delays in the evaluation of your manuscript. Please be aware that the editor who handles your revised manuscript might find it necessary to invite additional reviewers to assess this work once the revised manuscript is submitted. However, we will aim to proceed on the basis of this single review if possible.

The reviewer has provided a detailed assessment of your manuscript below. Please make the appropriate revisions to your manuscript in order to address their concerns.

• A letter that responds to each point raised by the editor and reviewer(s). You should upload this letter as a separate file labeled 'Response to Reviewers'.

We look forward to receiving your revised manuscript.

Kind regards,

Katrien G. Janin, PhD

Staff Editor

Journal Requirements:

i. Please clarify all sources of financial support for your study. List the grants, grant numbers, and organizations that funded your study, including funding received from your institution. Please note that suppliers of material support, including research materials, should be recognized in the Acknowledgements section rather than in the Financial Disclosure.

ii. State the initials, alongside each funding source, of each author to receive each grant. For example: "This work was supported by the National Institutes of Health (####### to AM; ###### to CJ) and the National Science Foundation (###### to AM)."

iii. State what role the funders took in the study. If the funders had no role in your study, please state: “The funders had no role in study design, data collection and analysis, decision to publish, or preparation of the manuscript.”

iv. If any authors received a salary from any of your funders, please state which authors and which funders.

If you did not receive any funding for this study, please simply state: “The authors received no specific funding for this work.

2. Please ensure that your Ethics Statement is available in its entirety at the beginning of your Methods section, under a subheading 'Ethics Statement'.

4. We note that there is identifying data in the Supporting Information file ‘Cleaned Dataset_Scabies_v3.xlsx’. Due to the inclusion of these potentially identifying data, we have removed this file from your file inventory. Prior to sharing human research participant data, authors should consult with an ethics committee to ensure data are shared in accordance with participant consent and all applicable local laws.

-Location data

Additional Editor Comments (if provided):

Reviewers' comments:

Reviewer's Responses to Questions

**Comments to the Author**

1. Does this manuscript meet PLOS Global Public Health’s publication criteria? Is the manuscript technically sound, and do the data support the conclusions? The manuscript must describe methodologically and ethically rigorous research with conclusions that are appropriately drawn based on the data presented.? Is the manuscript technically sound, and do the data support the conclusions? The manuscript must describe methodologically and ethically rigorous research with conclusions that are appropriately drawn based on the data presented.

Reviewer #1: Yes

2. Has the statistical analysis been performed appropriately and rigorously?

Reviewer #1: Yes

3. Have the authors made all data underlying the findings in their manuscript fully available (please refer to the Data Availability Statement at the start of the manuscript PDF file)?

The PLOS Data policy requires authors to make all data underlying the findings described in their manuscript fully available without restriction, with rare exception. The data should be provided as part of the manuscript or its supporting information, or deposited to a public repository. For example, in addition to summary statistics, the data points behind means, medians and variance measures should be available. If there are restrictions on publicly sharing data—e.g. participant privacy or use of data from a third party—those must be specified.requires authors to make all data underlying the findings described in their manuscript fully available without restriction, with rare exception. The data should be provided as part of the manuscript or its supporting information, or deposited to a public repository. For example, in addition to summary statistics, the data points behind means, medians and variance measures should be available. If there are restrictions on publicly sharing data—e.g. participant privacy or use of data from a third party—those must be specified.

Reviewer #1: No

4. Is the manuscript presented in an intelligible fashion and written in standard English?

Reviewer #1: Yes

Reviewer #1: 1. The WHO MDA program administered in 2023 should have been described in the background, including its extent (e.g., how many persons received MDA; what was coverage).

2. Looking at table 1, it seems that the surge occurred only from 2022-2023, and thus the costing for those two years can be averaged and compared to the pre-surge (2021) and post-surge/post-WHO MDA (2024)

3. For Table 2-Pls give how many grams of the permethrin 5% ointment per tube and what dosage frequency they were administered (single dose or 2 weekly doses?). also how many ml was flucloxacillin syrup, how many mg was cetirizine tab and syrup; Pls indicate currency of unit cost per drug; Pls provide costs for IPC supplies

4. What software was used to compute for costing?

Sentences in lines 138 and 139 are duplicates

Table 2 is not cited

Table 1 - pls provide total consultations for each year, which served as denominator to get the proportional morbidity

Vancouver style is not followed in some references

Ref 2 - only 1st letter of 1st word should be capitalized)

Ref#6 has unnecessary symbols (stars) after title

Ref #14-16 have no date of citation

Ref 4 and 8 should not have editor's names

**Do you want your identity to be public for this peer review?** For information about this choice, including consent withdrawal, please see our Privacy Policy..

Reviewer #1: **Yes:** Rowena GenuinoRowena GenuinoRowena GenuinoRowena Genuino

---

## [Decision Letter · Decision Letter 1]

11 Mar 2026

PGPH-D-25-03807R1

Economic Impact of a Large-Scale Scabies Upsurge on Healthcare Facilities in Rohingya Refugee Camps: A Retrospective Costing Study

Dear Dr. Halder,

Thank you for submitting your manuscript to PLOS Global Public Health. After careful consideration, we feel that it has merit but does not fully meet PLOS Global Public Health’s publication criteria as it currently stands. Therefore, we invite you to submit a revised version of the manuscript that addresses the points raised during the review process.

Please see the attached document and comments below regarding additional revisions required.

• A letter that responds to each point raised by the editor and reviewer(s). You should upload this letter as a separate file labeled 'Response to Reviewers'.

We look forward to receiving your revised manuscript.

Kind regards,

Claire J Standley

Academic Editor

Journal Requirements:

Additional Editor Comments (if provided):

Thank you for the revisions to the manuscript. Please see the attached file for additional grammatical and editorial revisions required. Please see also queries (particularly in the Methods) over treatment types, definition of complicated cases, and whether secondary bacterial infections were factored into the cost analysis.

Reviewers' comments:

Reviewer's Responses to Questions

**Comments to the Author**

Reviewer #1: All comments have been addressed

publication criteria? Is the manuscript technically sound, and do the data support the conclusions? The manuscript must describe methodologically and ethically rigorous research with conclusions that are appropriately drawn based on the data presented.? Is the manuscript technically sound, and do the data support the conclusions? The manuscript must describe methodologically and ethically rigorous research with conclusions that are appropriately drawn based on the data presented.

Reviewer #1: Yes

3. Has the statistical analysis been performed appropriately and rigorously?

Reviewer #1: N/A

4. Have the authors made all data underlying the findings in their manuscript fully available (please refer to the Data Availability Statement at the start of the manuscript PDF file)?

The PLOS Data policy requires authors to make all data underlying the findings described in their manuscript fully available without restriction, with rare exception. The data should be provided as part of the manuscript or its supporting information, or deposited to a public repository. For example, in addition to summary statistics, the data points behind means, medians and variance measures should be available. If there are restrictions on publicly sharing data—e.g. participant privacy or use of data from a third party—those must be specified.requires authors to make all data underlying the findings described in their manuscript fully available without restriction, with rare exception. The data should be provided as part of the manuscript or its supporting information, or deposited to a public repository. For example, in addition to summary statistics, the data points behind means, medians and variance measures should be available. If there are restrictions on publicly sharing data—e.g. participant privacy or use of data from a third party—those must be specified.

Reviewer #1: Yes

5. Is the manuscript presented in an intelligible fashion and written in standard English?

Reviewer #1: Yes

Reviewer #1: (No Response)

**Do you want your identity to be public for this peer review?** For information about this choice, including consent withdrawal, please see our Privacy Policy..

Reviewer #1: **Yes:** Rowena F. GenuinoRowena F. GenuinoRowena F. GenuinoRowena F. Genuino

---

## [Editor Report · Decision Letter 2]

18 Mar 2026

Economic Impact of a Large-Scale Scabies Upsurge on Healthcare Facilities in Rohingya Refugee Camps: A Retrospective Costing Study

PGPH-D-25-03807R2

Dear Dr. Halder,

We are pleased to inform you that your manuscript 'Economic Impact of a Large-Scale Scabies Upsurge on Healthcare Facilities in Rohingya Refugee Camps: A Retrospective Costing Study' has been provisionally accepted for publication in PLOS Global Public Health.

Best regards,

Claire J Standley

Academic Editor